



# Climatic and Societal Impacts of Volcanic Eruptions in the Western Han Dynasty (206 BCE- 8 CE): A Comparative Study

Zhen Yang[1] and Francis Ludlow[1]

[1]Trinity Centre for Environmental Humanities, Department of History, School of Histories and Humanities, Trinity College Dublin, D02 PN40, Ireland

*Correspondence to*: Zhen Yang (zhyang@tcd.ie)

**Abstract.** The Western Han dynasty (206 BCE-8 CE) experienced periods of extreme climate, accompanied by the evolving concept of the "mandate of heaven" that shaped societal response to disasters. While recent studies suggest that certain climate anomalies during the Han era are related to the atmospheric impacts of explosive volcanic eruptions, this paper employs both quantitative and qualitative methods to establish these associations more systematically. It categorizes and quantifies climatic stressors and selected societal events and applies superposed epoch analysis to examine the timing and statistical significance of their potential associations with ice-core-based dates of explosive eruptions. The paper then selects two historical periods, 180-150 BCE and 60-30 BCE, and offers a comparative analysis of recorded climatic and societal stresses, atmospheric optical anomalies, and societal responses to consecutive natural disasters. These periods are chosen because of the occurrence of massive volcanic eruptions known from polar ice-cores. For instance, in 43 BCE, when the Okmok volcano in Alaska erupted, a pale-blue sun and extreme summer cold are documented. Similarly, ice-cores identify a cluster of substantial eruptions in 168 BCE, 164 BCE, 161 BCE and 158 BCE that may have heavily impacted societies such as Egypt. Comparing the responses to the disasters of these periods also allow us to incorporate historical materials not suitable for quantification and to evaluate the effectiveness of Han dynasty disaster prevention and mitigation measures, thereby identifying factors that may contribute to better resilience to sudden and drastic environmental changes.

## 1 Introduction

He went out East Gate,

no hope to return;

he came in the gate,

he was shaken with grief.

No food in the kettle;

no clothes on the rack.

He drew his sword,

he went out the gate,

his children wept and wife pulled at his clothes.

"Other wives want wealth and honour,

I gladly share gruel with you,

share gruel with you:

By broad Heaven above,

by our babies here below,

this is wrong!"

"Get out of my way! I go!

I've waited far too long!

Already my hair hangs white,



I cannot stay here forever!"[1]

This conversation from "Parting and Going off, East Gate" (東門行), a classic piece of Yuefu (folk song style poem) possibly dated to the Eastern Han period (Owen, 1996), portrays how the hardship of lower-class livelihoods could promote societal instability. Given the poem's historical background, repeated disasters were likely one plausible trigger to the

husband's taking up arms in the face of the family's poverty. Environmental change is long considered a potential (if controversial) contributor to societal stress and collapse (e.g., Butzer, 2012; deMenocal, 2001; Fei and Zhou, 2016; Weiss and Bradley,2001; Haldon et al., 2020). Recent work suggests that climatic shocks have been persistently associated with Chinese dynastic collapse (or transition), with large shocks potentially acting more independently, and smaller shocks more often effective when dynasties were already experiencing notable stress (Gao et al., 2021). These shocks were inferred from

ice-core-based evidence for explosive volcanic eruptions, which are a known agent of abrupt regional to global climatic change. Our paper stages its research in the Western Han Dynasty (206 BCE- 8CE) to examine whether climatic and societal stresses recorded in Chinese sources are associated with explosive volcanic eruptions. As stated by Butzer (2012), academics have often overlooked "the intricate interplay of environmental, political, and sociocultural resilience" that can allow societies to rebound after impacts from climatic changes. We thus also explore how Han society responded to consecutive

disasters.

The short-lived but sudden and severe climatic impacts potentially wrought by explosive eruptions start with the sulfur-rich gases they can inject into the stratosphere, which there oxidize to form sulfate aerosols that backscatter incoming shortwave radiation, resulting in net surface cooling (Ludlow and Manning, 2021; Singh et al., 2023; Robock and Mao,1992). This is

direct radiative impact of explosive volcanism, but dynamical impacts are also regionally important. For tropical eruptions, the resulting concentration of aerosols in the stratosphere at lower latitudes (where they can induce high-altitude heating by absorbing outgoing long-wave radiation) may induce a north-south heating asymmetry that resolves itself partly by altering atmospheric circulation (e.g., enhancing westerly winter-time airflow in the Northern Hemispheric (NH) high latitudes (Robock and Mao,1992;Robock,2000; Zambri et al., 2017, Singh et al., 2023)). Extra-tropical eruptions, such as those in the

NH higher latitudes, are usually considered less impactful, but recent research suggests they can cause strong hemispheric cooling (Toohey et al., 2019). Radiative cooling from explosive volcanism can also induce precipitation anomalies that result in net global drying (though varying by region). One way is by reducing evaporation from waterbodies, reducing later precipitation potential (Iles et al., 2013). Another mechanism (particularly effective when eruptions occur in higher latitudes) is by restricting the summer season migration of the inter-tropical convergence zone into the hemisphere most heavily loaded

with aerosols, thereby diminishing associated monsoon rainfall (Robock 2020, Peterson et al., 2000; Fasullo et al., 2019). Volcanic aerosols and other particulates can also produce atmospheric optical anomalies that include diminished sunlight, a discoloured solar disk, unusually red "volcanic" sunsets, Bishop's Rings or coronae (Deirmendjian, 1973; Ludlow and Manning, 2021; Guillet et al., 2020). Such phenomena are documented through time across Eurasia (Scuderi, 1990; Wittmann and Xu,1987; Sigl et al., 2015).


Volcanically induced hydroclimatic extremes, especially cold and drought, can promote agricultural pest populations, land degradation, crop damage or bad harvests and human and animal mortality (Gao et al., 2021; Robock, 2020). Human mortality following major volcanic eruptions include diseases and high mortality in England and France after the Icelandic Laki eruption in 1783-1784 CE (Stothers,1999), the surge in diarrhoea cases in Bima, Dompo, and Sang'ir after the 1815 CE

---

[1] Translated by Stephen Owen. The original text is, 出東門，不顧歸。來入門，悵欲悲。盎中無鬥米儲，還視架上無懸衣。拔劍東門去，舍中兒母牽衣啼："他家但願富貴，賤妾與君共哺糜。上用倉浪天故，下當用此黃口兒。今非！" "咄！行！吾去爲遲！白髮時下難久居。



Tambora eruption (Oppenheimer, 2003), and the regional measles outbreak after the Mt. Pinatubo eruption in 1991 CE
      Philippines (Floret et al., 2006); all merit further examination regarding the mechanisms of disease spread after volcanic
      eruptions (Fei et al., 2016). Because some sulfate ultimately settles out of the atmosphere over polar regions and is captured
      in ice cores (Sigl et al., 2015), the climate forcing potential of past explosive volcanism can be investigated. Other archives
      such as tree-rings and historical documents can then allow any climatic and societal impacts to be identified and understood
(Yun et al., 2023; Oppenheimer, 2011).

      Although China's well-documented climate history provides a solid starting point to investigate impacts, few scholars have
      attempted this systematically. Exceptions are Gao et al. (2021), who identify a persistent interactive role for short-term
      climatic shocks (inferred from ice-core-based dates of explosive volcanism) and pre-existing societal stress (inferred from
warfare frequencies) in dynastic collapse during the Common Era. Specific studies of post-eruption cooling include the circa
      626 CE event (Fei et al., 2007), the Icelandic Eldgjá eruption in circa 939 CE (Fei et al., 2003), and the 1600 CE
      Huaynaputina eruption in Peru (Fei et al., 2016). The 1815 Tambora (Indonesia) eruption is most studied regarding the
      impact of sudden volcanic cooling, regional crop failure, famine and economic crisis in China (e.g., Yang et al., 2005; Zhang
      et al., 1992; Fei, 2018). Associations between drought and explosive volcanism (Shen et al., 2007), and solar phenomena and
eruptions (Zhang, 2007), have also been examined. Furthermore, recent studies have increasingly referenced climatic and
      atmospheric optical records in Chinese sources as a complement to natural archives, for instance, research on the 43 BCE
      Okmok (Alaska) eruption (McConnell et al., 2020; Wang et al., 2024), the 852/3 CE Mount Churchill (Alaska) eruption
      (Mackay et al., 2022), and other medieval-era eruptions (Guillet et al., 2023). However, the comparatively abundant
      historical climate records of China hold much further potential for this purpose.


      The Han Dynasty (206 BCE- 220 CE) is known for frequent natural disasters and it's ever-evolving "ominous politics", with
      these two elements together profoundly shaping the society. Much research has thus been dedicated to understanding the
      contemporary interpretation of climate anomalies as "portents" endowed with cultural and political agency (e.g.,
      Bielenstein,1950; Eberhard,1933; Cai, 2015). Our paper concentrates on the Western Han Dynasty as it experienced multiple
volcanic eruptions, with two periods particularly notable in having sharply increased polar sulfate deposition around 44 BCE
      and 168 BCE (Sigl et al., 2015; Toohey and Sigl, 2017). Tree-ring growth anomalies and reconstructed summer temperature
      corroborate the ice-cores in suggesting the 160s BCE and 40s BCE were extremely cold (Sigl et al., 2015). Because sulfate
      deposition was identified in both poles, the 168 BCE eruption is considered tropical in location. This was followed by three
      smaller but notable extratropical NH eruptions in 164 BCE, 161 BCE and 158 BCE (deemed such because their sulphate was
confined to Greenland (Sigl et al., 2015), with potentially corresponding records from Babylon (Iraq) of dim suns in 164
      BCE and 161 BCE (Sachs and Hunger,1996; McGovern, 2024), hydroclimatic anomalies and unrest in Ptolemaic Egypt
      (Ludlow and Manning 2021, Singh et al., 2023). Chinese records of a discoloured sun and cold temperatures around 43 BCE
      have already been noticed and linked to a documented 44 BCE eruption of Etna (Sicily) (e.g., Bicknell 1993, Fei et al.,
      2016). A more recent reassessment has convincingly associated these phenomena (and relevant Greenland sulfate) with a
massive 43 BCE eruption of Okmok (Alaska), closely following a notable extratropical NH eruption in 46 BCE (McConnell
      et al., 2020).

      Scholars often structure their analyses of Han history according to each emperors' reign; this includes climatic history given
      the political importance of disasters, materially and as omens. The 160s BCE fell under the rule of Emperor Wen, while the
40s BCE were in Emperor Yuan's reign, which encountered the most disasters, especially water-related disasters,
      earthquakes, snowfall and famines, of all the Western Han dynasty (Jiao et al., 2009). These events leave a considerable
      imprint on the dynasty's record and we aim to identify the climatic stresses, atmospheric optical anomalies and societal





impacts and responses that appear statistically associated with eruptions during the dynasty's duration. This is facilitated by applying Superposed Epoch Analyses to variables including warfare and vagrancy derived from the historical sources. A

qualitative comparison will then focus on two three-decade periods comprising the years before, during and after the volcanic episodes of 168-158 BCE and the 40s BCE to allow a closer examination of attempts to restore political, economic and societal stability following multiple potentially volcanically induced climatic and societal impacts.

## 2 Material and Methods

### 2.1 Survey and categorization of documented climate anomalies and impacts

To create a comprehensive chronological dataset of climatic records for the period, historical sources are re-surveyed and their content integrated with established datasets that include *A Compendium of Meteorological Records of China in the Last 3000 Years* by Zhang et al. (2004), *The General History of Natural Disasters in China-the Volume of Qin and Han* by Jiao et al. (2009), and the *Table of Natural and Human Disasters in Chinese Dynasties* by Chen et al. (1933), as well as two studies on Han climate that provide lists of relevant records (Chen, 2016; Chen, 2001). These all slightly differ regarding the events

included. Of primary sources, dynastic histories are closely surveyed, especially *Shiji* (史記, *Records of the Grand Historian*, by Sima Qian, completed 91 BC) and *Hanshu* (漢書, *Book of Han*, by Ban Gu, completed 82 CE). Within *Hanshu*, climatic and atmospheric optical phenomena are mainly found in the *Annals, Treatise on the Five Elements* and *Treatise on Astronomy*, but are also scattered in some of its constituent biographies. Thus, it was deemed necessary to survey all volumes of *Hanshu*, together with *Shiji* and other relevant sources. These include *Zizhi Tongjian* (資治通鑑,

*Comprehensive mirror to aid in government*, by Sima Guang, a chronicle recording Chinese history from 403 BC to 959 CE) and *Wenxian Tongkao* (文獻通考, a comprehensive examination of literature from earlier periods, compiled by Ma Duanlin in 1307 CE).

Similarly, new chronological datasets for harvest conditions, warfare (including rebellions), population changes (including

that arising from vagrancy, the movement of garrisons and planned migration) and amnesties (pardons to the convicted on special occasions, sometimes including awards to the common people) have been derived as potential proxies for societal stress and mechanisms of response. The new warfare dataset builds upon the *Tabulation of Wars in Ancient China* (Editing Committee of China's Military History, 1985), complemented by a thorough reading and lexical search of *Hanshu*. The dataset of amnesties builds on Wu's (2003) study of the Han dynasty's amnesty system. No pre-existing compilations of Han

harvest conditions or population change exist, although Ge's (2014) study of Western Han population and geography offers a solid foundation for the later theme. Hence, the records in these two datasets (see supplementary material) are extracted from hundreds of annals, accounts, tables, treatises and biographies of *Hanshu* and *Shiji*. Only direct and credible attestations to harvest conditions and population changes are included. All the datasets mentioned above are provided in the Supplement of this paper.

### 2.2 Time-series and quantification of climatic and societal impacts

The reliability of each record in our datasets is assessed by applying general principles of historical and textual criticism (e.g., McNeil, 2000) adapted to the context of the Han era. Within each dataset, records are categorized to construct annual frequency series for meteorological and related extremes. When documented disasters are ambiguous regarding location or extent, these are taken as applying to the northern and central part of the empire, where the major population and the major

agricultural areas were located. Decadal frequency time series of all variables are also provided for a perspective on broader trends. However, these frequencies cannot capture all relevant information in the available records. Thus, we also present an



"impact index", building upon Chen (2016), and capitalizing on three common features of our climate records that state (or imply) different dimensions of the scale of events: phrasing, duration and affected area. Each feature is assigned a score from 1-3 (for wording, from mild to serious, then extreme; for duration, from days to months and then years; for affected area,

from local to regional, then empire-wide). Multiplying these scores produces an impact index, with higher scores equating to greater impact. These are provided for the most serious climate disaster of each given year. For more nuance, we also create individual time-series of events plausibly representing impacts arising (at least in part) from (or occurring in response to) climatic shocks that are documented in sufficient abundance to attempt statistical analysis. These include years of bad harvests, annual famine and plague incidence, annual warfare and rebellion frequencies, years experiencing vagrancy and

annual planned migration incidence.

### 2.3 Superposed epoch analysis

Superposed epoch analysis (SEA) is widely employed to overlay and average (i.e., composite or superpose) multiple temporal sections of a given time-series (of a potentially dependent phenomenon) around a series of specifically dated focal events known or hypothesized to have a direct (or indirect) causal influence on (or correlative association with) the

potentially dependent phenomenon (e.g., Manning et al., 2017; Rao et al., 2019; Campbell and Ludlow, 2020; Gao et al., 2020). This allows an assessment of whether consistent responses register in the potentially dependent phenomenon, during or following the focal event years. SEA has been widely used to assess hydroclimatic responses to volcanic eruptions (e.g., Rao et al., 2019) and has been increasingly adapted to examine potential post-volcanic responses in societal variables (e.g., political revolts in Ptolemaic Egypt (Manning et al., 2017) or Chinese dynastic collapses (Gao et al., 2020)). Our focal events

are years when ice-core data (Sigl et al., 2015) imply explosive eruptions, whilst the new climatic and societal variables (described above) are the dependent or response timeseries. Only tropical and northern hemisphere extratropical eruptions are considered, in being most likely induce adverse climate in the Western Han empire. Fourteen such eruption years are identified from 206 BCE to 8 CE, comprising 190, 180, 168, 164, 161, 158, 147, 141, 123, 105, 85, 46, 43 and 35 BCE.

### 2.4 Comparative case studies

The Western Han's 214-year span enables a closer comparative analysis of two episodes of explosive volcanism already recognized in Ptolemaic Egypt and the Ancient Near East for inducing climatic-societal impacts, namely, the 168-158 BCE "volcanic quartet" and the 40s BCE (McConnell et al., 2020; Ludlow and Manning, 2021; Kostick et al., 2023; Singh et al., 2023). Our qualitative case studies of the historical periods 180–150 BCE and 60–30 BCE complement the SEA approach by allowing consideration of records less suitable for quantitative analysis, particularly regarding societal responses to climatic

stress evolving on different timescales. The case study approach allows us to ask, moreover, whether responses changed between episodes. The 40s BCE certainly inherited a political legacy from the 168–158 BCE period, but significant changes had also occurred in state-governance, disaster perception, and more. In each case study, responses are also contextualised by reference to climatic and societal conditions in the decade immediately preceding and following the eruptions, and attention paid to whether responses were effective in impact mitigation.

## 3. Results and Discussion

### 3.1 Quantitative Analysis

#### 3.1.1 Climatic impacts and atmospheric optical anomalies

Figure 1 (top panel) shows the diversity of documented climate-induced (flood, drought, cold) and potentially related (famine, plague, locusts) disasters and other societal stressors (warfare and conflict) in decadal totals. The figure also

presents combined counts of disasters across categories (with and without famine and plague) plus decadal impact index





scores. Annual disaster frequencies are shown in the bottom panel, with years experiencing one or more disasters being common. Coherent trends of genuine climatic origin playing out on multi-decadal scales are difficult to confidently identify, however; particularly given the many potential non-climatic influences on disaster incidence and their recording (e.g., Ludlow, 2012). Nonetheless, some years and decades apparently suffered notably more frequent and impactful disasters. The

impact index for our case studies reveals the 40s and 30s to be particularly impacted, whereas the 160s and 150s BCE exhibit more moderate (if certainly not low) scores. The high scores for the 40s and 30s BCE result from increased counts across all disaster categories (excepting locusts), a marked contrast to the lows of the 50s BCE. The annual frequency data shows that disasters in 163 and 158 BCE are the main contributors to the elevated decadal values of the 160s and 150s BCE, whereas those in 48 and 43 BCE make the most substantial contribution to that decade. It may be notable that of these individual

years, 158 and 43 BCE coincide with ice-core-based dates of major eruptions, whereas 163 BCE follows by just one year. The possibility cannot be excluded, however, that such close timings are coincidental. We thus employ an SEA approach to assess (1) whether eruption dates more broadly co-occurred with (or were followed by) increased disasters during the Western Han era more broadly and (2) whether any increase arose by chance.

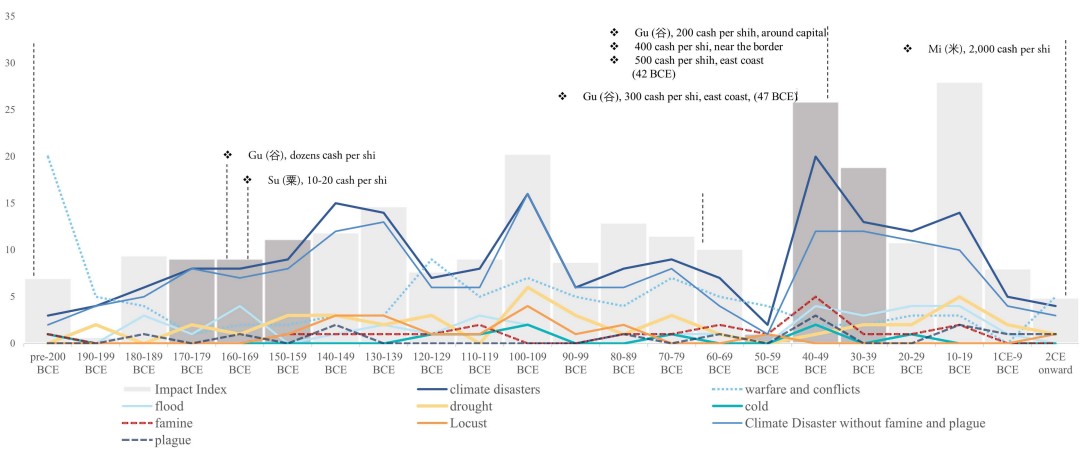


**Figure 1. The top panel shows the time-series of various proxies of climatic and societal stresses in the Western Han dynasty at a decadal resolution. The bottom panel presents the time-series of the number of climate disasters, both including and excluding instances of famine and plague, at an annual resolution.**





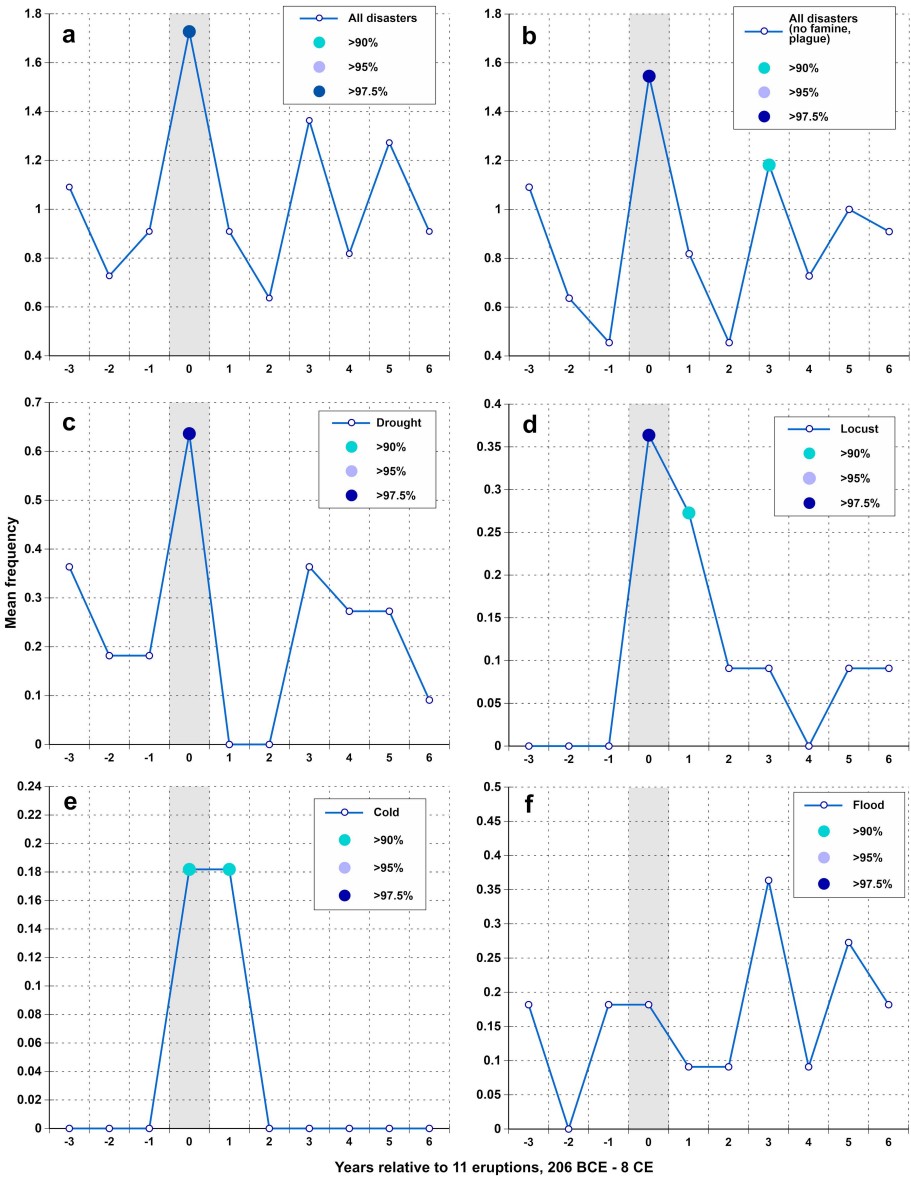

**Figure 2. Superposed Epoch Analysis (SEA) results of the time series of climate disasters relative to 11 tropical and Northern Hemisphere (NH) volcanic eruptions (identified based on ice-core evidence) as focal event years**

Figure 2 thus visualizes multiple SEA outcomes in which our disaster time-series are superposed relative to the tropical and extratropical NH eruption dates taken from Sigl et al. (2015) but adjusting the Okmok eruption date from 44 to 43 BCE, following McConnell et al. (2020). Our analyses employ ten-year superposition windows that show (as a reference baseline) the mean frequencies for our disaster categories in each of the three years preceding eruptions (Superposed Years -3 to -1, horizontal axes, Fig. 2a-f), the mean frequency in the eruption years themselves (Superposed Year 0), and the mean in each of the six years following (to identify any potential multi-year or lagged post-eruption impacts (Superposed Years 1 to 6)). We note that three of the 14 eruption dates are omitted because their inclusion introduces a "double counting" artefact that can occur in the SEA approach when focal event dates recur closely in time and are comparatively small in number



(Manning et al., 2017). In addressing this issue (which manifests in our analyses as an artificial secondary peak in Superposed Year 3) we identify eruption dates that occur within three years of each other and exclude the smaller of these events (based on the Greenland sulfate deposition volumes of Sigl et al. (2015)) as being less likely to elicit a climatic and societal response. We thus omit (1) 164 BCE, which follows the larger eruption of 168 BCE, (2) 161 BCE, which precedes the larger eruption of 158 BCE, and (3) 46 BCE, which precedes the larger eruption of 43 BCE.

Taking Fig. 2a, which considers the combined climate and related disaster time-series (comprising 198 events, distributed across 113 years of the 214-year Western Han era, and ranging from zero to a maximum of six in any given year), we observe the highest peak in mean disaster frequency during eruption years. While this suggests a volcanic influence, we must also assess the likelihood it has arisen randomly. To do this, we follow the approach of Campbell and Ludlow (2020) and Gao et al. (2021) in randomly re-ordering the combined climate and related disaster time-series 10,000 times, on each randomisation calculating the mean frequency of the randomly re-ordered disasters that fall in each of the years (-3 through 0 to plus 6) relative to our 11 set eruption dates. This produces a randomised reference distribution against which to compare the mean values observed relative to our eruption dates when using the actual (original) time series of climate and related disasters. The outcome is shown in Fig. 2a, where any actually observed superposed means (blue line, Fig. 2a) that fall within the top 10%, 5%, or 2.5% of means in our random reference distribution are colour-coded and taken, respectively, as having a less than 10%, 5%, or 2.5% likelihood of occurring purely by chance (i.e., they are considered statistically significant at > 90%, >95%, or >97.5% confidence). Ultimately, the mean disaster frequency in Year 0 (the eruption years themselves) is the only value deemed statistically significantly elevated (at 98.4% confidence), supporting the identification of a recurrent role for explosive volcanism in triggering climate and related disasters during the Western Han era.

While our records leave little doubt that famine and plague were often associated by contemporaries with adverse climate conditions, it is known that they may also have multiple non-climatic origins (e.g., population movement, conflict) (Gao et al., 2016). We thus repeat the above SEA but remove disasters solely involving these events from the combined time-series (leaving 164 events across 107 years, ranging from zero to four in any given year). The highest peak nonetheless persists in Year 0 (Fig. 2b) and remains statistically significant (now at 99.4% confidence). A smaller secondary peak in Year 3 now registers as marginally significant (at 90.8%).

It is also instructive to examine the association between volcanic dates and disaster categories individually. Figure 2c thus shows that mean drought frequencies (43 events across 39 years, ranging from zero to two in any given year) peak suddenly in Year 0 (at 99.6% confidence and represent the only statistically significantly elevated year), while locust disasters (18 events across 16 years, ranging from zero to two in any given year) shown in Fig. 2d peak in Years 0 and 1 (at 98.7% and 93.3% confidence, respectively). No other years approach significance in either category. Cold disasters (Fig. 2e) show equivalent peaks in Years 0 and 1 (each significant at 90.6% confidence), suggesting a multi-year temperature impact consistent with expectations (e.g., Gao et al., 2017). Even so, the lower significance of cold disasters may be surprising given that cold is totemic of volcanic climatic impacts (at least for summer, as some NH continental landmass regions may experience transient winter warming (Robock and Mao, 1992; Robock, 2000; Zambri et al., 2017)). We note, however, only nine cold disasters are recorded for the Western Han dynasty (distributed across eight years, with seven registering one such disaster and only one year experiencing two), with the small sample size likely bearing upon these results. Of all disaster categories, flooding (36 events across 33 years, ranging from zero to two in any given year) alone fails to exhibit any notably elevated frequencies (Fig. 2f). This is not unexpected given our contrasting finding of increased drought frequency in the same broad area, northern and central China, which accords with findings of increased drought in China in other periods following tropical and NH extratropical eruptions (Zhuo et al., 2014; Gao et al., 2017).



### 3.1.2 Societal Impacts

To assess whether societal stressors exhibit any notable volcanic association, we begin by superposing our annual impact index values (Section 2.2) on our 11 volcanic focal dates (Fig. 3a). Index values of between one and 27 are assigned across 114 years, with all other years assigned a value of zero. We see the mean index value peak in Year 0 (statistically significant at 98.2% confidence), before falling below the 90% threshold, but remaining elevated relative to pre-eruption values. This is unsurprising given the elevated mean disaster frequencies seen in Section 3.1.1. For individual stressors, we start with bad

harvests, of which 26 such years are identified. Their mean superposed frequency indeed peaks in Year 0 (Fig. 3b), but (at 87.4%) does not meet the minimum 90% confidence threshold. This may be partly explicable given that we consider these events less consistently reported in the sources. In addition, the variance captured in the time-series itself is limited to presence-absence (0,1) format. We cannot, therefore, at present identify whether the poor harvests associated with eruptions were particularly widespread or severe.


For famine and plague, we are more confident that the sources capture most events of any notable scale. However, the respective frequencies remain small, with 21 famines across 21 years (thus ranging between zero and one in any given year) and 13 plagues across 11 years (mainly ranging between zero and one in any given year, with just one year experiencing two plagues). In Fig. 3c, we observe a moderately elevated mean famine incidence in Years -1, 0, 2 and 3, though none reach the

90% confidence. In Fig. 3d, the only statistically significant peak in mean plague frequencies is observed for Year -1 (96.6% confidence), with no plague documented in any of our 11 eruption years. A one year offset could be plausibly attributed to small uncertainties in ice-core layer counting and/or small delays between eruption dates and the deposition of sulphate in the ice (Sigl et al., 2015; Gao et al., 2016). Under this interpretation, the plague peak in Year -1 might be associated with our volcanic eruptions. However, this is inconsistent with the recurrent peaks in climate disasters that do occur in Year 0 (Fig. 2)

as well as the impact index peak in Year 0 (Fig. 3a). While our randomisation testing is intended to mitigate some drawbacks of small number statistics, they cannot be entirely circumvented. It is thus notable that the peak in Year -1 depends entirely on one plague reported in 44 BCE (one year before the 43 BCE Okmok eruption) and one in 142 BCE (one year before the unknown 141 BCE eruption). Future work may address these limitations (e.g., by creating harvest, famine and plague indexes that derive greater information from the available evidence concerning spatial extent or severity that may be

particularly associated with volcanically triggered cases).

For cases of vagrancy, in which many people are documented as abandoning their homes or land, often in search of food, we identify 33 years experiencing this phenomenon and our time-series is presence-absence (0,1) in format. Figure 3e reveals elevated mean frequencies in Years 2 and 3 (statistically significant at 93.2% and 92.9%, respectively), suggesting a

somewhat delayed onset of this behaviour in response to adverse conditions wrought by eruptions. In cases of planned migrations, the state deliberately relocated large numbers of people for many potential reasons, such as temporary labour needed for construction, or to create garrisons, but also plausibly to increase food security by returning abandoned land to productivity or facilitate expansion into uncultivated territories (precedents for which exist elsewhere in the ancient world (see Ludlow et al., 2023)). We identify 44 years experiencing such migrations (again presence-absence in format) and see

that their mean frequency increases suggestively in the three years following our eruptions (Fig. 3f), though never reaching statistical significance (Year 3 comes closest at 83.9%).

Finally, we examine the potential influence of volcanically induced climatic shock on violence and conflict, starting with annual warfare and rebellion frequencies (139 events across 90 years, ranging from zero to 11 in any given year). The

scholarly literature on climate-conflict linkages is undecided on whether such events may increase or decrease following climatic stress (see Section 3.2.2), and we thus test for the statistical significance of movements in both directions. Notable

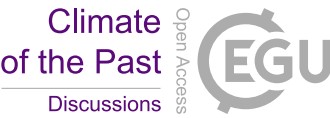

in Fig. 3g, therefore, is an unusually low mean frequency in Year 1 (reaching statistical significance, at 98.8% confidence) that follows from another low in Year 0 (though not itself significant, at 88.3% confidence). Figure 3h repeats the test, but considers warfare only (i.e., 101 events across 71 years, after omitting internal rebellions, and ranging from zero to eight in any given year). Much the same pattern is observed here, with Year 1 being the sole value to reach statistical significance (at 98.9%). When considering rebellion only (66 events across 50 years, ranging from zero to four in any given year), there are apparently modest frequencies in Years 0 and 1, but these do not emerge as statistically significantly high or low. These results are further discussed in Section 3.2.2.



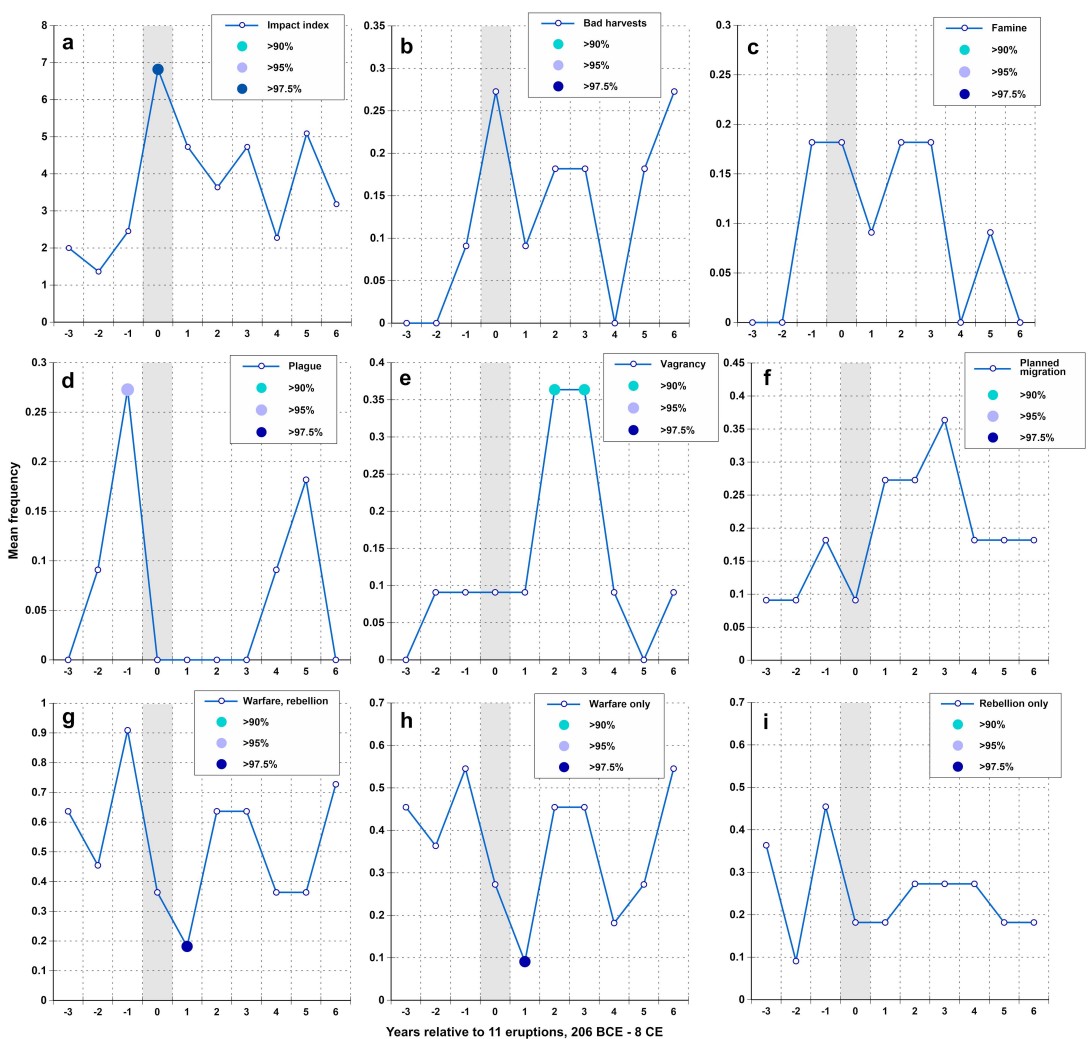

**Figure 3.** Superposed Epoch Analysis (SEA) results of the time series of societal stress proxies relative to 11 tropical and Northern Hemisphere (NH) volcanic eruptions (identified based on ice-core evidence) as focal event years



### 3.2 Comparative Case Study

**3.2.1 Climatic impacts and atmospheric optical anomalies**

Climate records of the Western Han dynasty (Table S1 in the Supplement) suggest that the 170s BCE, which did not experience any known explosive eruptions (Sigl et al., 2015), suffered only occasional short-term local or regional flooding, snowfall and windiness. Two state-wide droughts in autumn 177 BCE and spring 171 BCE may, however, have contributed some mounting climatic stress before the "volcanic quartet" of four sequential eruptions in 168, 164, 161 and 158 BCE. In

the 160s BCE, the Yellow River first overflowed in the winter of 168 BCE in today's Henan province, then flooded again in 165 BCE, importantly highlighting that post-volcanic hydroclimatic conditions can depart from expectations (e.g., of eruptions associated with drought, as per Section 3.1.1). We may infer that these challenging circumstances continued, with an imperial edict in 163 BCE noting that:

"Recently for many years there have continually been no good harvests. Moreover, there have been visitations of floods, droughts, sickness, and epidemics…" (*Hanshu, Annals of Emperor Wen*; Dubs, 1938). [詔曰："間者數年比不登，又有水旱疾疫之災……" 《漢書・文帝紀》]

In autumn 161 BCE, continuous rain is reported to have caused severe floods that submerged thousands of houses and killed

hundreds of people. By contrast, the 150s BCE was a decade of serious drought, more consistent with post-volcanic expectations, even if a causal link cannot be drawn conclusively between specific single eruptions and droughts. A drought from spring 158 BCE lasted into summer and led to a locust plague, with drought visiting again in 155 BCE, afflicting the central and northwest of the empire.

Of our second case study, most years in the 50s BCE experienced almost no natural disasters (Table S1). Only one local locust outbreak is recorded from 60-49 BCE. This situation reversed dramatically in the 40s BCE, however. Even before the eruptions of 46 and 43 BCE, large-scale severe flooding in 48 BCE contributed to famines, epidemics and even cannibalism in Guandong (east of Hangu Pass), where most population and farmlands were concentrated. The situation further deteriorated through the decade and the evidence of 43 BCE clearly implies a volcanic contribution. The annals, treatises,

and multiple biographies in *Hanshu* (Table S1) thus document a pale-blue sun, with the *Treatise on the Five Elements* providing the most detailed description:

"In the 4th month (May 7-June 5), the sun was pale blue (blue-white) in colour and cast no shadows. Right in the middle (of the Sun) frequently there were shadows and no brightness. That summer was cold until the 9th month, the

Sun then regained its brightness…" (*Hanshu, Treatise on the Five Elements*; Yau and Stephenson,1988) [元帝永光元年四月，日色青白，亡景，正中時有景亡光。是夏寒，至九月，日乃有光。《漢書・五行志》]

The pale-blue sun was accompanied by extreme summer cold, marked by frosts in the 3rd (April 8- May 6) and 9[th] (approximately October) months. While the former damaged wheat and mulberries, the latter effectively killed the year's

harvest and famine swept the whole empire in 42 BCE, possibly triggering societal unrest (Section 3.2.2). In the 30s BCE, after an initial cross-regional flood and overflow of the Yellow River (in 39 BCE), disasters occurred less frequently and severely. The incidence and timing of the pale-blue sun as well as the extremity of conditions that same year (and that immediately following) can be credibly associated with the heavy atmospheric aerosol loading and consequent radiative cooling expected from a massive eruption like Okmok in 43 BCE (McConnell et al., 2020).





### 3.2.2 Societal Impacts


Our first case study period (180-150 BCE) witnessed one plague and famine. The plague occurred in 163 BCE, notably just one year after the 164 BCE eruption, but can also be considered a combined outcome of the multiple disasters from previous years (Section 3.2.1). The famine occurred in 156 BCE, just two years after the 158 BCE eruption and a potentially associated drought and locust plague (discussed above), and was certainly serious:


… an imperial edict said, "Recently, for successive years there have not been good harvests, so that many people are lacking food; early death is cutting short their natural [span of] years…" (*Hanshu, Annals of Emperor Jing*; Dubs, 1938) [詔曰：" 間者歲比不登，民多乏食，夭絕天年……" 《漢書・景帝紀》]


In comparison, the 40s BCE suffered almost continuous famine. The empire-wide famine from October 43 BCE caused grain prices to rocket, as documented in 42 BCE:


"At the time, bad harvests had lasted for years. Each *Shi* (a measurement unit, approximately 30 kg today) of grains cost more than 200 cash in the capital, but about 400 per *shi* around the remote areas, and about 500 per Shi in Guandong. There were famines everywhere, and the imperial court was very concerned about the situation, and it also encountered mutinies of the Qiang people (an ethnic group)." *Hanshu, the Biography of Feng Fengshi* [是時，歲比不登，京師谷石二百餘，邊郡四百，關東五百，四方饑饉，朝廷方以爲憂，而遭羌變。《漢書・馮奉世傳》]


For context, Han grain prices during 180-150 BCE varied within a more constrained range of dozens to 100 cash per shi (Huang, 2002; Tan, 1994), though we have no prices explicitly recorded for any of the volcanic quartet years in order to assess any short-term departures from this price range. After a period of commodity price increase due to the prevalence of counterfeit coins around 120 BCE (Ban and Swann 2013), grain prices restabilized during a period of good harvests, reaching a low of 5 cash per shi in 62 BCE *(Hanshu, Annals of Emperor Xuan)*. Prices were thus dramatically higher in 42


BCE, outstripping the already high price of 300 per shi in 47 BCE *(Hanshu, Treatise on Food and Money)*. Unfortunately, there is no information regarding the 30s BCE. These price changes should be considered in tandem with evidence of the harvest situation (Table S4 in the Supplement). The first bad harvest during 180-150 BCE is recorded for 163 BCE, representing the first of an apparent run of poor harvests that lasted at least into 156 BCE. The timing of these ongoing harvest failures is almost parallel with the three volcanic eruptions of 164, 161 and 158 BCE, yet the years after 168 BCE


(with its large tropical eruption) have documented bumper harvests, thereby exemplifying the complexity of volcanic climatic impacts. Additionally, whereas the bad harvests of the 40s BCE are reflected by marked grain price fluctuation, the poor yields of the 160s and 150s BCE did affect prices enough as to merit their inclusion in the surviving records.

Disasters are often linked to large-scale internal population movements (e.g., Chumky et al., 2022), and population


movements can be inferred from records of vagrancy for our first case study period in 178, 174 and 156 BCE, with planned migrations (mostly for apparent administrative and agricultural reasons) in 178, 177, 170, 169, 165 and 152 BCE (Table S5 in the Supplement). The latter thus do not align well with the dates of the volcanic quartet eruptions (unless the migration in 165 BCE is considered as a potential part outcome of the 168 BCE eruption). The available records do, however, document that vagrancy in 156 BCE was indeed a consequence of bad harvest and famine (see this record in table S5). This may be


plausibly connected to the 158 BCE eruption, with some additional possible role for cumulative impacts of earlier quartet eruptions. Planned migrations also occurred in our 60-30 BCE case study period, with those in the years 59, 57, 55 and 48



BCE all pre-dating known eruptions, and only the planned migration of 42 BCE plausibly associated (chronologically) with explosive volcanism (the 43 BCE Okmok eruption). The sources suggest that these initiatives were undertaken mainly for military purposes (e.g., garrisoning and settling surrenderers). Vagrancy was certainly frequent, occurring in 48, 47, 43, 42 and 32 BCE, and often in explicit association with the ongoing disasters of the 40s BCE (with a plausible volcanic role for events in 43 and 42 BCE).

Assessing any role for climate in warfare and rebellion is challenging given the multiple "pathways" that might act either synergistically or antagonistically to promote or suppress violence and conflict, with the net outcome in any instance mediated by cultural, political, and environmental contexts and the specific violence and conflict typologies considered (Ludlow et al., 2023). A commonly cited pathway by which violence and conflict might rise after climate and related disasters involves competition for reduced resources (e.g., food after harvest failure) (Homer-Dixon, 1999). The operation of such a pathway can be plausibly linked to known challenges in our period such as vagrancy. This might compound food supply issues (e.g., if agricultural land is abandoned), reduce social stability (e.g., via tensions between incumbent and newly mobile groups seeking subsistence, potentially through theft), and place the dynasty under financial strain with lost revenues from land abandonment at a time when expenditure needs rise to support the population. We might also posit an increased need for military intervention to deter or suppress rival state activity aimed at exploiting Western Han difficulties, or external mobile non-state actors (e.g., pastoralists) opportunistically (or by necessity) raiding into Western Han territories, or internal but distinct groups hoping to cede from Western Han control.

Despite the above, the net general outcome for warfare and rebellion frequencies appears as a reduction one-year post-eruption (Fig. 3). For organised warfare and large-scale rebellion at least, the difficulties of conducting military operations during adverse conditions (Winters et al., 1998) or of resourcing troops when harvests had deteriorated and societal needs and potential internal instability was higher (e.g., Manning et al., 2017) may have won out (on net) over pathways promoting increased warfare or large-scale rebellion. Detailed examination reveals that most wars of the period were between Han and ethnic groups, or they were revolts led by vassal kings, marquises or tribes. Their overt causes appear political rather than climate or disaster driven. This is broadly consistent with most rebellions occurring in the pre-eruption decade of our 180-150 BCE case study, with none occurring during or within three years post-eruptions, except a relatively small-scale rebellion plotted by Xin Yuanping (新垣平) in 164 BCE. However, while the net general outcome of volcanically induced climatic disasters may have been to suppress warfare and rebellion, in specific cases the opposing pathways may still have prevailed, and it is notable that warfare between the Han government and the Xiongnu, a confederation of nomadic tribes, was recorded in 166, 162 and 158 BCE, close to or in eruption years. Whether these events can be solidly linked to climatic conditions merits further study.

In our 60-30 BCE period, repeated rebellions occurred in the Zhuya Commanderies (current Hainan) until its abolition in 46 BCE, during a time when the government faced repeated climate and related disasters. Instructive of competing state imperatives (and the antagonistic pathways mentioned above) is Jia Juanzhi's (賈捐之, an official) attempt to convince Emperor Yuan to withdraw rather than suppressing the insurgents and instead focus more on disaster relief in Guandong *(Hanshu, Biography of Jia Juanzhi)*. Similarly, the rebellion of the Qiang tribe in autumn 42 BCE occurred when sources suggest that the whole empire suffered bad harvests, extreme grain prices and famine. When General Feng Fengshi (馮奉世) requested 40,000 soldiers to address the revolt, other officials opposed his suggestion, citing that there was no extra manpower available since people were busy with the autumn harvest. Feng argued:





"…the whole state is suffering from famine. The soldiers and horses are emaciated, and they have been greatly consumed. The equipment for defence and combat has been disused for a long time. The barbarians all show contempt for our officials at the border. That is why the Qiang people revolt …" (*Hanshu, Biography of Feng Fengshi*) ［天下被饑饉，士馬羸秏，守戰之備久廢不簡，夷狄皆有輕邊吏之心，而羌首難。《漢書‧馮奉世傳》］

Ultimately, he received only 12,000 soldiers and eventually the emperor had to send larger forces (over 60,000 soldiers) to restore order. Thus, climatic and societal impacts potentially induced by the 43 BCE Okmok eruption may in this case have played important roles in initiating conflict and then influencing (e.g., constraining) state strategies taken in response. It is also worth noting that warfare was not the only form of violence and conflict reported in the sources. Increasing robbery and brigandage are reported in an edict in 42 BCE:


"…But the Yin and Yang have not yet accorded [with each other], the three luminaries have been veiled and indistinct, the great multitude have suffered greatly, have wandered, and have been scattered on the highways and paths. Robbers and brigands have arisen simultaneously…" (*Hanshu, Annals of Emperor Yuan*; Dubs 1938) ［……然而陰陽未調，三光晻昧。元元大困，流散道路，盜賊並興。《漢書‧元帝紀》］


In 41 BCE, a similar record is found, blaming an earthquake, rain and fog in mid-winter (Table S1). The possible statistical association between crime and volcanic climatic impacts merits further research.

### 3.2.3 Societal Responses

The immediate responses taken by the Western Han government in our case study periods share similarities but also
significant differences due mainly to changes in agricultural policies, disaster perception, and socio-economic conditions. A summary of these policy changes sees Fig. 4.

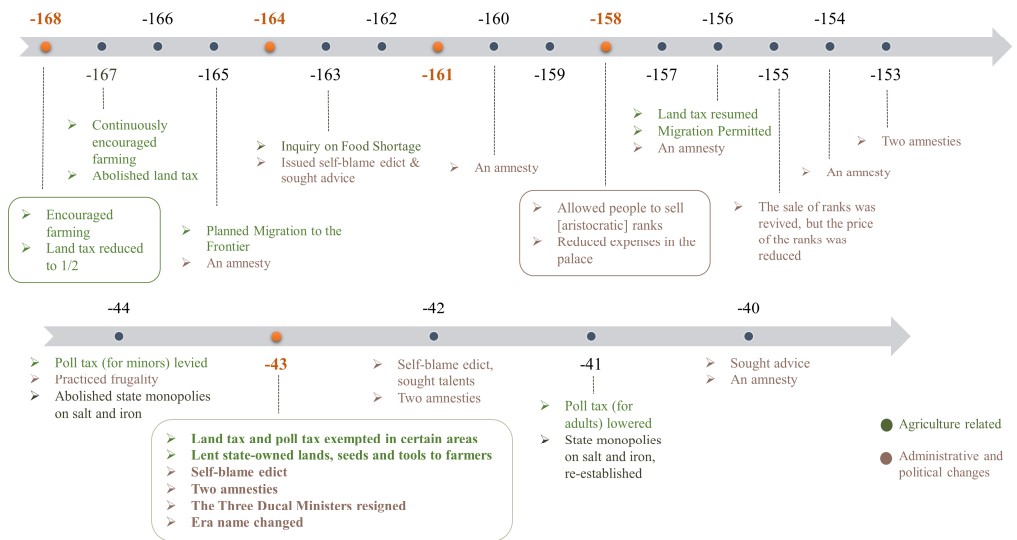

**Figure 4. Summary of important policy changes related to the case study period**



From 180-169 BCE, actions to address natural disaster impacts are barely documented, excepting a self-blame edict (a proclamation in which the emperor takes responsibility for disasters) in 178 BCE following a solar eclipse. This may imply that disaster relief was not too demanding in human and monetary terms in these years. For the Yellow River flooding of 168 BCE, however, extensive labour was assembled to repair the embankment *(Hanshu, vol.29)*. After further flooding of the river in 165 BCE there is no specific infrastructure damage recorded but the event was treated as an "omen" for which the

emperor issued orders to build a temple and offer sacrifices *(Shiji, Vol.28)*. The situation became more serious in the years leading up to 163 BCE, as the state suffered the combined consequence of crop failures, floods, droughts and diseases. A classic self-blame edict (the full text edicts in our two case study periods is given in tables S2 and S3) was issued by the emperor with a request to seek suggestions from officials, especially for the causes of food shortage. A further flood occurred in autumn 161 BCE, but responses are absent from the records. By contrast, five approaches were documented

when spring drought in 158 BCE lasted into summer and triggered locusts:

[The emperor] ordered that the nobles should not pay tribute. He opened [to the common people] the mountains and marshes, reduced the [imperial] robes and the imperial officers, diminished the [regular] number of Gentlemen and officials, and opened the granaries in order to succour the people. The people were allowed to sell [aristocratic] ranks.

(*Hanshu, Annals of Emperor Wen*; Dubs,1938) [令諸侯無入貢，弛山澤，減諸服御，損郎吏員，發倉庾以振民，民得賣爵。《漢書‧文帝紀》]

These five approaches comprise tribute reduction, opening access to state-owned natural resources, practicing frugality, opening granaries and selling ranks. While the first four were common strategies during climate disasters, the sale of noble

ranks was recorded more rarely as a measure to address disasters, for instance, the permission to sell ranks was reissued with reduced price in 155 BCE *(Shiji vol.30)* to address drought in the same year. Ranks in the Han period could be sold for cash or used as ransom for reducing punishment (Hsu, 1980). The Han government began the practice of trading ranks for grain with the intention of solving supply issues for armies stationed at borders while elevating the status of farmers, but it developed into a strategy to secure food during disasters in the reign of Emperor Wen and Jing (180 BC-141 BC). However,

this measure increased social differentiation (Li, 2006), as the rich could also buy ranks that entitled them to corvée exemption and immunity for past and even future crimes (Hsu, 1980). Migration was also permitted in 156 BCE to address food shortages, as the emperor recognized the challenges arising from an uneven distribution of population and agricultural sources and hoped to encourage farming in underdeveloped areas.

In our 60-30 BCE period, with climatic stresses accumulating rapidly from 48 BCE onward, all common relief measures were implemented (sending officials to investigate circumstances, seeking talents (i.e., to enlist the help of talented persons) and suggestions, tax and loan reductions or exemptions, opening access to state-owned resources, granaries and more). An edict in the 3rd month, likely before the first frost in 43 BCE documented in *Hanshu*, already granted an amnesty to allow the newly pardoned to *"improve their personalities, renew themselves, and each pay attention to cultivating his acres"*

(*Hanshu*; Dubs, 1938). The amnesty also included loaning fields, seeds and food to the amnestied and poor, and awarding ranks and other items like wine and silk to a variety of officials and people. However, actions more symbolic than practical appear to have been taken to address the continued summer cold and pale-blue sun, namely, the resignation of the Three Ducal Ministers' (thereby attributing them some blame for circumstances) and the change of era name from Chuyuan to Yongguang (永光), which implies eternal brightness. For the frost in the ninth month of 43 BCE and the great famine that

followed, no measure is documented until the next spring. As part of a self-blame edict (in the second month), an amnesty was granted accompanied by the giving of noble ranks and items to common people, gold to the nobles and silk to the virtuous especially the diligent cultivators of fields (*Hanshu, vol.9*). A month later, a solar eclipse precipitated another edict,





blaming the increasingly violent and cruel behaviour of the people as the source of the sun's diminished light, and again the emperor sought good and capable talents (*Hanshu vol.9*). The ongoing bad harvest led to a third edict (in the sixth month) in

42 BCE, together with the fourth amnesty since 43 BCE. Yet another edict and amnesty were issued that exempted the poor from loans and debts in 40 BCE, partly intended to address unrest. This makes self-blame edicts and amnesties the most documented state measures in this case, used more frequently than at other times in the Western Han.

Comparing self-blame edicts from our two case study periods reveals the differences arising from changing perceptions of

climate anomalies. Han philosopher Dong Zhongshu (董仲舒), the founder of Han Confucianism, conceptualized natural disasters as "warnings from heaven", which were initially used to constrain the ruler's power. However, his theory was taken further and became more influential later, especially during Emperor Yuan's reign, as he revered Confucianism. When Emperor Wen and Jing's edicts emphasized inquiry about causes of disasters and sought to tackle problems, Emperor Yuan encouraged officials to discuss moral issues and government policies by interpreting portents. However, the self-blame

edicts do not necessarily imply the emperor's willingness to take responsibility. In 43 BCE, Emperor Yuan, being criticized for his inability to trust the virtuous and eliminate the evil (Huang 2016), preferred to blame the Three Ducal Ministers, who were believed to be responsible for harmonizing Yin and Yang *(Hanshi Waizhuan; Hanshu)*. Eventually, Yu Dingguo (于定國), the Lieutenant Chancellor, Shi Gao (史高), the Commander-in-chief and Xue Guangde (薛廣德), the Grandee Secretary, all resigned. This was the first time in Western Han history that the three Ducal Ministers openly and proactively

took responsibility for their "strict liability" (Jiao et al 2009). Furthermore, the interpretation of meteorological anomalies was often contested and manipulated by rival elites as tools for partisan struggle. Indeed, the dimmed sun in 43 BCE was exploited by the powerful eunuchs Shi Xian (石顯) and Hong Gong (弘恭) to suggest the demotion of Zhou Kan (周堪) and Zhang Meng (張猛), their political antagonists. Interestingly, in 40 BCE, Zhou and Zhang came back to the court when the emperor interpreted a solar eclipse that year as a sign that he made a mistake in precipitating these resignations *(Hanshu,*

*vol.36)*. Thus, in practical terms, "moral cosmology" (Cai, 2015) as a concept parted ways from its original intention to constrain power or promote Confucius' ethical ideals. Furthermore, the political struggles (including over the interpretation of omens) that occurred during disasters arguably mad disaster relief less effective.

Similarly, amnesties usually involved freeing prisoners and could be permitted for various reasons, including as a celebration

of the enthronement of a new emperor, celebrating auspicious meteorological signs, and more. In the case of climate anomalies, the granting of amnesties was also a way to cultivate a virtuous society and thereby ward off future disasters (Jiao et al., 2009). Thus, most amnesties were granted to comfort the public rather than have substantial effects on disaster relief, and the accompanying awards such as cattle, wine, silk and noble ranks might serve the same purpose. Such measures could initially ease the tensions between different classes because many convicts were common people who struggled to survive

when the privileged class's benefits were barely affected by natural disasters. Loaning fields and seeds to the amnestied people and turning them into agricultural labour could also contribute to societal stability during disasters, but consecutive amnesties may have been of reduced effectiveness. As reported by an official in 42 BCE, those who had been pardoned repeatedly committed crimes and social behaviour was not improved *(Hanshu Vol.81)*. This may imply that the burden on the common people's livelihood was not effectively relieved during disasters.


From an economic perspective, the gap between rich and poor became increasingly significant during the Western Han era. The bigger this gap is, the more vulnerable society becomes to climate extremes. Contemporary documents, such as Jia Yi's (賈誼) memorial, stated *"Licentious and luxurious customs increase day by day",* while there were those who attempted to sell their children *(Hanshu vol.24, Swann 2013)*. Chao Cuo's (晁錯) memorial (in 178 BCE), mentioned that independent





farmers could not *"make both ends meet even in a year of normal crops"* and had to borrow at high-interest rates to pay
exorbitant taxes (*Hanshu vol.24*, Hsu 1980). In the 40 BCEs, the contrast became more obvious. In Gong Yu's (貢禹)
memorial (44 BCE), he compared the expenses in Emperor Wen and Jing's court to inform Emperor Yuan of the excessive
extravagance enjoyed by the privileged class and some government departments, while farmers continuously suffered from:

"…having paid a [regular] tax in kind of grain, and also having paid the hay tax, the private demands solicited by
authorities of the district and the village become too heavy a burden to bear" (*Hanshu vol.72*; Hsu, 1980)〔已奉谷租，
又出稿稅，鄉部私求，不可勝供。《漢書・食貨志》〕

To address social and economic inequalities, one common state strategy during disasters was practicing frugality, which was
undertaken in 158 BCE by reducing the number of imperial officers and four times in the 40s BCE by cutting imperial
expenses in banquets, entertainment and horses (48, 47, 44 BCE), reducing palace officers (46 BCE) and abolishing offices
(44 BCE). For the overtaxed poor, with the intention to encourage farming, land tax was reduced by half to 1/30 of the
annual produce in 168 BCE and was completely abolished for 12 years until it resumed to 1/30 in 156 BCE *(Hanshu)*. The
state also charged poll tax. While suanfu (算賦, poll tax on adults) was reduced three times to 40 cash during Emperor Wen's
(179-157 BCE) rule, with a gap in its documentation, it was next known to be 90 cash in 52 BCE and decreased to 80 cash in
41 BCE (Hsu 1980). As for kouqian (口錢, poll tax on minors), there are no records concerning this during our earlier case
study period, but sources show that its first taxable age was raised from three to seven in 44 BCE. This move was suggested
by Gong Yu, considering the difficulties of the people's livelihood (*Hanshu vol.72*). However, whether other changes in
taxes were related to disasters is not documented. There were indeed tribute exemptions (158 BCE) and land tax exemptions
in affected areas (48, 47 BCE) to offer immediate relief after disasters. Other economic changes during these two periods
include the "Ever-normal granary (常平倉)", established in 54 BCE as a state instrument for the leveraging of grain prices to
benefit farmers but this was abolished in 44 BCE, after several years of climatic disasters, when officials pointed out that it
should not *"strive* [i.e., compete] *with the people for profits"* (*Hanshu, vol. 24*). The state monopoly on salt and iron
production was disestablished in the same year but resumed in the winter of 41 BCE as the government had insufficient
income due to too much tax exemption *(Hanshu, vol.9)*. The association between these economic changes and possible
volcanic-induced climate stress requires further exploration, but the repeated disasters of the 40s BCE drained the empire's
wealth, further affecting the government's ability to implement disaster relief measures later.

As discussed, many policy changes in Han time before, during and after recurrent climate disasters aimed to encourage
agricultural activities (Fig. 4, table S2, S3 in the Supplement). However, during times when the harvest was doomed to be
damaged by extreme weather, such measures were hardly helpful. In Emperor Wen's time, agriculture was generally
underdeveloped therefore production was low and grain storage was insufficient (Hsu, 1980), regardless of good harvests.
Under this circumstance, it was intuitive that officials and the emperor prioritized the development of agriculture, even in
disaster prevention and relief. Nevertheless, as pointed out by Hsu (1980), one big problem of Han agriculture was that the
heavy burden of non-agricultural expenses and taxes exhausted farming resources, and eventually independent farmers lost
their lands and became tenants that were constantly exploited. The government neglected this aspect and blamed people for
overly pursuing commercially profitable activities therefore causing food shortages (e.g., the edict in 163 BCE, table S2),
indicating the ruler's misunderstanding of agricultural development. Hence, chronic food shortages went unresolved while
businesses were at times needlessly suppressed.



## 4 Conclusion

This paper has employed a mixed-method approach to explore associations between explosive volcanism and climatic stresses, societal impacts and potential responses in Western Han sources. Statistically, the mean frequency of combined climate and related disasters (plus individual categories such as drought, cold and locusts) exhibited statistically significant increases in association with volcanic eruption dates. A detailed qualitative examination of climate records from 180-150 BCE reveals that the 170s BCE was relatively quiescent regarding natural disasters, while the 160s BCE experienced recurrent floods, with plague and poor harvests in 163 BCE potentially associated with explosive volcanism in 164 BCE, and large-scale extreme droughts in the 150s BCE potentially associated with explosive volcanism in 158 BCE. In our second case study period, the years 60-49 BCE barely experienced disaster, but the climatic stresses suddenly mounted in 48 BCE and continued over the following years of the decade. Although we cannot conclusively attribute all these events to volcanism, the hydroclimatic extremes during and shortly after the volcanic quartet years of 168 to 158 BCE, and the sudden extremes of cold and atmospheric optical anomalies in 43 BCE are consistent with expectations of the atmospheric and climatic consequences of major explosive eruptions.

Of societal impacts, our impact index and individual metrics such as vagrancy demonstrate statistically significant associations with volcanic eruptions, whilst other metrics should as the incidence of bad harvests and famine are suggestive in their association, if not formally statistically significant. From a qualitative perspective, there were poor harvests from 163 BCE to the mid-150s BCE and again in the 40s BCE that can be plausibly associated with eruptions, and the latter caused rapid grain price increase. For conflict, we observed that warfare frequencies statistically significantly decreased after volcanic eruptions, with our case studies showing that the parties involved in warfare could experience pressure to refrain and direct their resources to relief efforts during disasters.

Our results show that Western Han society and government experienced ongoing vulnerability to periodic disasters, but also demonstrated considerable resilience in persisting in the face of these events. Nonetheless, gaps between awareness and knowledge, as well as shortcomings in the perception and understanding of the environment and human-nature relationships, restricted the development of its disaster prevention and mitigation system. As an agrarian society, it relied heavily on the reliability of weather. However, the "warnings from heaven" theory, conceived as a constraint to power, was instrumentalized for political struggles among the bureaucrats, who abused their posts to gain wealth, while the lower classes suffered. In addition, the Han government realized the importance of food storage for disaster prevention, as mentioned in Jia's memorial, *"...increases the stores of grains in order to fill government granaries and roofed depots, in preparedness against floods and droughts..."* (*Hanshu vol.24*; Ban and Swann, 2013) 〔"……廣畜積，以實倉廩，備水旱……"《漢書‧食貨志》〕, but strategies employed to achieve this goal could be ineffective or counterproductive. For instance, the sale of ranks for the government to collect food later aggravated social differentiation and inequality; the "ever-normal granary" for leveraging grain prices was abolished as it competed with the people for profits and possibly because the government could not afford its high-maintenance during ongoing disasters. Similarly, in disaster relief, tax reduction or exemption might initially offer assistance, but later drain the state's wealth and hinder further disaster relief. All of these, together with strategies to encourage farming by restraining the development of business, resulted in a less diverse (and thereby potentially more fragile) economy. These developments decreased society's resilience to natural disasters and may be considered examples of maladaptation, comprising intentional adaptation measures that eventually increase society's vulnerability (Juhola et al., 2016).

In general, the reigns of Emperor Wen and Jing (180-141 BCE) are known for their steady development, while Emperor Yuan (48-33 BCE) is associated with Western Han decay. However, we cannot conclude that the earlier period had more



effective systems and measures to address disasters. Firstly, the severity and frequency of climatic anomalies varied in these two periods. The latter suffered more devastating consecutive disasters, which left the society limited recovery time.

Secondly, changing perceptions linked to "ominous politics" might affect the documentation of climatic anomalies. There could be more records during the less peaceful time, especially with a ruler like Emperor Yuan who highly praised Confucius classics and encouraged criticism (Huang 2016). Thus, we can only examine what history has left to us, and compare it with other evidence, and that requires more innovative exploration with both quantitative and qualitative methods. The other limitation is the relatively short period we study. This makes possible a comprehensive reading of the

available historical sources but may affect the statistical power of SEA approaches by limiting sample sizes. Thus, extending the period under study (e.g., including the period from the Qin to Eastern Han dynasties (221 BCE - 220 CE)) offers one way forward. Finally, applying contemporary theories and frameworks to evaluate the effectiveness of disaster relief measures may not be suitable for such an early dynasty. However, one important criterion is always the individuals' livelihood, which calls for the lens of micro-history. With the gradually uncovered but still limited historical materials and archaeological

evidence of the Han era, literary works, such as the poem in the beginning provide a glance into this aspect and offer new research opportunities.

**Data availability**

The chronological dataset of climatic records is provided in Table S1. Major policy changes and their relevant translations are listed in Table S2 (180–150 BCE) and Table S3 (60–30 BCE). The chronological datasets of harvest conditions and

population changes in the Western Han Dynasty are presented in Table S4 and Table S5, respectively. All tables are provided as supplementary material for this paper. Volcanic eruption events identified through ice-core data can be found in Supplementary Data 5 of https://doi.org/10.1038/nature14565 (Sigl et al., 2015).

**Author contributions**

ZY and FL conceptualized and designed this paper. ZY conducted the research, analyzed historical records, and constructed

the datasets. ZY and FL performed the superposed epoch analysis (SEA), and FL produced the graphs that visualized the SEA results. ZY prepared the original draft of the paper, and FL was involved in the editing process.

**Competing Interests**

The authors declare that they have no conflict of interest.

**Financial support**

Zhen Yang and Francis Ludlow (project PI) received funding from the ERC Synergy 4-Oceans project (Grant agreement ID: 951649). Francis Ludlow acknowledges additional support from NSF CNH-L award number 1824770.

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
