# Peer review of "Climatic and Societal Impacts of Volcanic Eruptions in the Western Han Dynasty (206 BCE- 8 CE): A Comparative Study"

_Climate of the Past, 2024_

## Author Comment (AC1)

**Responses to Reviewer 1**

**R 1.1:** This manuscript presents a comprehensive study on the impacts of volcanic eruptions on climate and society during the Western Han Dynasty (206 BCE - 8 CE). The authors employ a combination of quantitative and qualitative methods to systematically link volcanic activity with climatic and societal stressors. By categorizing and quantifying climatic stressors and societal events, and utilizing superposed epoch analysis, the study reveals statistically significant associations between volcanic eruptions and increased frequencies of climate-related disasters. Additionally, the paper provides a comparative analysis of two specific periods marked by massive volcanic eruptions, showcasing different societal responses and the potential effectiveness of the Han dynasty's disaster mitigation measures. This research enhances our understanding of how past societies coped with environmental catastrophes and underscores the importance of historical records in assessing societal resilience to sudden environmental changes. The study is well-structured, with a clear methodology and robust analysis, offering valuable insights into the interplay between natural disasters and societal dynamics in ancient China.

**A 1.1:** We appreciate the reviewer's positive feedback and constructive comments.

Specific comments:

**R1.2:** Line 119: It is recommended to provide a brief introduction to Emperor Wen of Han, such as "the fifth Emperor of the Han Dynasty," in parentheses. Similarly, several Chinese historical terms are not explained upon their first appearance. Examples include "Yin and Yang" in line 466 and "Gong Yu" in line 566. It is advised to review the entire text and offer brief explanations of these ancient Chinese terms when they first appear.

**A1.2:** We have carefully reviewed our manuscript and provided brief introductions for Chinese historical terms in brackets after their first appearance, where necessary. For instance, in line 119, Emperor Wen of Han (the fifth Emperor of the Han Dynasty); in line 466, "Yin and Yang" (a Chinese philosophical concept regarding balance and harmony between opposites); in line 567, Gong Yu (124-44 BCE, a high official), etc. As historians, we can sometimes overlook the need to provide terminological explanations for multi-disciplinary audiences. We appreciate this recommendation, which helps us improve the manuscript's accessibility.

**R1.3:** Line 130-142: This paragraph only describes the original sources of climate-related descriptions. However, the reader would also like to know which types of climate-related records (e.g., frost, snow, thunder, drought, flood, etc.) have been extracted from these historical documents. It is also recommended to present this information in table form in section 2.1, detailing the types of data extracted from historical documents and the number of records for each type.

**A 1.3:** Detailed information on the types of climate-related records extracted from historical documents is provided in column D, "Summary," in supplementary table S1. To improve clarity in the main article text, we have added a note in this section to direct readers to reference the relevant supplementary tables. We have also (as suggested) added a table in the main text exemplifying the types of data extracted from historical sources and the number of records per type.

**R1.4:** Line 149: The manuscript mentions that there were no pre-existing compilations of Han harvest conditions. However, there are indeed studies on the harvest of the Han Dynasty. The following literature is provided for reference.
[1] Yin J , Su Y , Fang X .Relationships between temperature change and grain harvest fluctuations in China from 210 BC to 1910 AD[J].Quaternary International, 2015, 355:153-163.

[2] Yun S , Xiuqi F , Jun Y .Impact of climate change on fluctuations of grain harvests in China from the Western Han Dynasty to the Five Dynasties (206 BC–960 AD)[J].Science China Earth Sciences, 2014(07):1701-1712.

[3] Fang X Q, Su Y, Yin J, et al. 2015. Transmission of climate change impacts from temperature change to grain harvests, famines and peasant uprisings in the historical China. Science China: Earth Sciences, 2015,58(8):1427-1439.

**A1.4:** Thank you for highlighting these references. While we were aware of these articles, a new chronology of annual figures was needed to facilitate our Superposed Epoch Analysis (SEA). The suggested papers present their data primarily in graphs and summary tables without a corresponding annual chronology, and thus we were unable to immediately incorporate them directly into our analysis. Moreover, we decided to create a chronology based only upon direct attestations of poor harvests (appropriate for SEA) without inferring additional poor harvests (e.g., from reported flooding or insect outbreaks). This differentiates our approach from existing studies. Nonetheless, we have happily added these references to our manuscript to acknowledge their contributions and have revised our phrasing to clarify that the absence of pre-existing compilations refers specifically to those comprising only explicit annual attestations of poor harvests complied to facilitate the SEA approach. In our planned future papers on climate and harvest conditions over a longer period of imperial China, we will also cite and discuss this valuable pre-existing work in more detail.

**R1.5:** Line 159-160: It is suggested that the territory of the Han Dynasty, along with major areas and important place names (such as Hangu Pass) mentioned in the text, should be illustrated here.

**A1.5:** Thank you for the suggestion. We will add a map to illustrate the major areas and important place names. However, this may depend on whether there is available data for open-access publication and its suitability for the reuse and reproduction policies of *Climate of the Past*. If suitable data is not available, we will provide explanations of the territory, areas, and place names instead.

**R 1.6:** Line 200: It is necessary to explain here (similarly to the expression in lines 252-253) why disasters are categorized into two groups: those with famine and plague, and those without.

**A1.6:** Thank you for highlighting this ambiguity. It is known that famine and plague can be triggered by climatically induced shocks to agriculture and food supply (e.g., harvest failures and animal mortality promoting subsistence crises that can compromise human immune systems). Mass movement of people in search of subsistence can also facilitate the spread of disease. However, in both cases, the underlying societal context is critical, and indeed both famine and plague can occur in the absence of any meaningful climatic input. For example, conflict can impact food supply through scorched earth warfare (targeting crops and animals), by removing agricultural labour supply (when diverted to military service), or indirectly impact food availability by increasing market prices when food is requisitioned for military supplies or trade routes are disrupted. This diminishment of food supply and availability can again impact human immune systems, while mass movements of peoples as part of military service or to escape conflict can similarly facilitate disease spread (these mechanisms are listed in Gao et al. (2021)).

Ultimately, because our SEA testing was conducted to examine the potential volcanic role in climate and climate-related disasters, we decided to test whether the apparent link between documented disasters and volcanic eruptions persisted both with and without the inclusion of famine and plague-related disasters, given that these latter disasters can also have non-climatic origins, as discussed above. As reported in the main article text, we found that disaster frequencies increased in both cases. We have now made the rationale for this decision clearer in the manuscript.

**R 1.7:** Figure 1 (top panel): I have the following two suggestions.

(1) The figure is not clear enough. It is recommended to mark the meaning of the ordinate on the graph. Since most values in the figure are below 25, it is advisable to set the value range of the ordinate to 0-25 and enlarge the smallest unit to show the differences among different indicators more clearly.

(2) The annotations in the figure need further explanation. For instance, Gu (谷), Su (粟), and Mi (米) should be identified according to modern taxonomy. Additionally, clarify the unit "shi" and provide its conversion relationship with the current international unit.

**A 1.7:** Thank you for the suggestions; we have adjusted the figure accordingly.

**R 1.8:** Line 383-395: According to Line 383-385, the grain price ranged from 200 (in the capital) to 500 (in Guandong) in 42 BCE, while the grain price was 300 in 47 BCE according to line 395. It is unclear how the conclusion was reached that grain prices were higher in 42 BCE than in 47 BCE.

**A 1.8:** Thank you for noticing this. The information regarding the grain price of 300 cash per shi in 47 BCE is originally from *Hanshu, Treatise on Food and Money*, and refers specifically to the Guandong area. The entirety of the relevant text is:

"When the emperor Yuan (48 BCE) came to the throne, there was a great flood in the empire which was especially calamitous in eleven commanderies [and kingdoms] to the east of the [Hangu] Pass. In the second year, crops in the territory of Qi failed. Grains [cost] more than 300 cash per shi. The greater part of the inhabitants starved to death. In Langya [Commandery], residents ate human flesh." [This translation references Ban and Swann (2013), with slight adjustments to ensure the spelling and terms are consistent within this manuscript. The original text is: 元帝即位，天下大水，關東郡十一尤甚。二年，齊地飢，穀石三百餘，民多餓死，瑯邪郡人相食。]

Hence, to improve clarity, we have adjusted the relevant text in our manuscript to: "Prices were thus dramatically higher across various regions in 42 BCE, with those in the Guandong area in particular outstripping the already high price of 300 per shi in 47 BCE".

**R 1.9:** Line 431-443: The interpretation of the net general outcome for warfare and rebellion frequencies as a reduction one-year post-eruption is debatable. While I agree with the authors that revolts led by vassal kings, marquises, or rebellions may be caused by political reasons, many studies argue that one of the significant reasons for conflicts between agricultural civilizations and nomadic civilizations is climate change. As mentioned in lines 440-443, there are indeed records of wars between the Han government and the Xiongnu following the volcanic eruption period. It is suggested that this issue be discussed in greater detail or that the relevant wording in this paragraph be modified. The following literature is provided for reference.

[1] Pei Q, Lee H F, Zhang D D, et al. Climate change, state capacity and nomad–agriculturalist conflicts in Chinese history[J]. Quaternary International, 2019, 508: 36-42.

[2] Su Y, Liu L, Fang X Q, et al. The relationship between climate change and wars waged between nomadic and farming groups from the Western Han Dynasty to the Tang Dynasty period[J]. Climate of the Past, 2016, 12(1): 137-150.

**A 1.9:** Thank you for this discussion and these specific references. We were aware of studies attributing conflicts between sedentary agricultural societies and nomadic peoples to climatic pressures, and do not dispute this general finding, including for China and proximate peoples. However, there are several

considerations when interpreting the net general outcome for warfare and rebellion frequencies in our SEA analysis, in relation to findings from previous studies.

Firstly, our research, especially the SEA results discussed in lines 431-443, focuses only on the statistical association between warfare and explosive volcanic eruptions (as discrete events), rather than warfare and the broader spectrum of possibly conflict-relevant climatic disasters (or temperature and precipitation variability as a continuous variable). In addition, we have focused on the Western Han (206 BCE -8 CE), which is a more constrained period than some previous studies that have observed an increase in conflicts between nomadic or pastoralist peoples and the mainly sedentary agriculturalist populations of Chinese dynasties through time. Moreover, our warfare chronology includes a broader selection of conflicts (e.g., inter-state war and internal rebellions) instead of only nomadic/pastoralist conflicts with Western Han sedentary/agriculturalist populations.

We posit that these considerations can explain the apparent lack of (potentially expected) statistically significantly elevated warfare frequencies in or following the years of explosive volcanic eruptions during the Western Han era. Further, we suggest that the statistically significantly reduced warfare frequencies that are by contrast observed in the first superposed year following eruptions can be potentially explained by the increased cost of prosecuting large-scale (including inter-state) war amid climatic and associated agroeconomic adversity. This is consistent with findings of inter-state warfare cessation between Ptolemaic Egypt and the Seleucid Empire following eruptions (Manning et al., 2017).

However, because our SEA analysis reveals only the *average* warfare frequency response in and following volcanic eruption years, our results should not be interpreted as implying that eruptions in the Western Han era never had *any* promotional effect on conflicts. This may have occurred for specific individual cases, or on particular types of conflicts, such as those with sedentary agricultural societies and nomadic peoples, but not sufficiently to drive up the overall average to a statistically significant value. Ultimately, a study of explosive volcanic eruptions over a longer period with a clear differentiation between different conflict typologies is warranted to tease out the nuances of climate-conflict linkages through time, space and by type. We have accordingly revised the main text to improve clarity on this issue in the manner discussed above and will add the suggested citations.

Thus, we hope the discussion above explains why, although some warfare (especially Han-Xiongnu conflicts) took place in the post-volcanic eruption periods, throughout the entire Western Han Dynasty, elevated warfare frequencies did not on average synchronize with volcanic eruptions. Such an outcome may not hold when a longer period is included, but again, the current research outcome does not in principle conflict with the findings of existing studies.

**R1.10:** Figure 4: It is recommended to add a style annotation for volcanic eruption dates to the legend.

**A1.10:** Thank you for the suggestion. The years marked in orange indicate volcanic eruption dates. We have added them to the figure legend to improve clarity.

**R1.11**: Line 605-661: The last few paragraphs of the conclusion chapter seem more like content suited for the discussion chapter. It is recommended to change the chapter title to "Conclusion and Discussion" or to divide the conclusion and discussion into two separate chapters.

**A1.11:** We have adjusted the discussion and conclusion sections addressing this comment and also reviewer 2's relevant comments.

**R1.12:** Supplements: There are errors in the translation and interpretation of historical documents, especially in the Table S1. Here are a few examples.

(1) Line 52: "蓝田山" is not the name of a specific mountain, but generally refer to a mountain in Lantian county.

(2) Line 129, 145 and 170: "Xiongnu" is the name of a nomadic people, not a place name.

(3) Line 149: "Weiqiao" (渭桥)：means a bridge over the Wei River and cannot be used as a location name. Similarly, "commanderies and kingdoms" ("郡国"in line 166, 174, 177 and 198) cannot be used as a location name, either.

**A1.12:** We thank the reviewer for reading the supplementary tables so carefully. We originally used "Xiongnu" and "commanderies and kingdoms" to approximately indicate affected areas. However, placing such information under the "location" column was indeed not precise enough. We have now made corrections accordingly.

For instance, we now clarify that the entries referring to the Xiongnu indicate the affected areas where the relevant groups of nomadic people usually resided, which is approximately in present-day northern China. This information has been moved to the "area" column. In addition, "藍田" in this entry refers to Lantian County, and we have corrected it accordingly. "Weiqiao" ("渭橋") has been changed to "the Wei [River] Bridge" (Ban and Dubs, 1938), with an explanation that the term is often used to imply the approximate capital area, and it has been placed under the "area" column.

We have also re-examined all historical terms in the supplementary tables and have made necessary changes or added explanations accordingly.

**R1.13:** Technical corrections:
Line 179: The reference here should be Gao et., 2021.

**A 1.13:** This has been corrected.

---

## Author Comment (AC2)

**Responses to Reviewer 2**

**R 2.1:** The manuscript titled "Climatic and Societal Impacts of Volcanic Eruptions in the Western Han Dynasty (206 BCE - 8 CE): A Comparative Study" provides a comprehensive case investigation of the potential climatic and societal impacts of volcanic eruptions on ancient China, employing both quantitative and qualitative analysis. The study is well designed, and the results help foster our understanding of the interplay between the nature-induced-disasters and human society beyond the common era. I therefore recommend publication of the study after addressing the following comments and issues raised by other reviewers.

**A 2.1:** We thank the reviewer for their supportive comments.

**R 2.2:** Section 2.2, in addition to list the data sources, please describe how the data (for example, the ones in Figure 1) are compiled. How are the frequencies accounted? How are the time series reconstructed, which data sources/chronicles have been included in different periods? Are they consistent, especially during the two periods of 180-150 BCE and 60-30 BCE?

**A2.2:** Thank you for these queries. The method for compiling data is provided in Sections 2.1 and 2.2. Each entry included in our frequency count is listed in the supplementary tables, with a column clarifying the source and reference for each record.

As outlined in section 2.1, our approach involved a thorough re-surveying of available historical sources and the integration of their content with established datasets including *A Compendium of Meteorological Records of China in the Last 3000 Years* by Zhang et al. (2004), *The General History of Natural Disasters in China-the Volume of Qin and Han* by Jiao et al. (2009), and the *Table of Natural and Human Disasters in Chinese Dynasties* by Chen et al. (1933), as well as two studies on Han climate that provide lists of relevant records (Chen, 2016; Chen, 2001).

*Hanshu*, the dynastic history, is ultimately our main historical source, along with *Shiji*. If the same event is documented in both sources, we counted it only once to avoid repeated counting. Thus, in terms of research method and source, our counting is consistent throughout the whole period under study — the Western Han Dynasty — including the case study years 180-150 BCE and 60-30 BCE. In addition, we compared our counts with all the existing datasets mentioned above. In rare cases, where there are slight differences, we re-examined the historical contexts and details of the relevant records to confirm our decision about whether the event in question should be counted. This has also been detailed in the notes of Table S1 and the additional "Notes for the Tables" document in supplementary materials.

To improve clarity in the main article text, we have revised the relevant wording and added additional explanation to address this comment (as well as the related comment R1.3 of the first reviewer, in response to which we have added a table to the main text exemplifying the types of data extracted from our historical sources and the number of records for each disaster type).

**R 2.3:** Figure 1. It would be nice to see the information about the 11 eruptions, for example, the cumulative radiative forcing of the eruptions, in this figure. Please also state in the figure caption, where the data drawn can be found (tables in the Supplemental Information, etc).

**A 2.3:** We thank the reviewer for these suggestions.

We have now added a new figure to the manuscript. Ultimately all our volcanic dates and forcing data come from the study of Sigl et al. (2015), which provides the relevant data in its supplementary files. However, there are many ways by which to graph the number and forcing potential of the volcanic events under study for our period. Since the data originally come in the form of ice-core sulphate

deposition measurements, we have plotted the annual chronology of sulphate deposition (in ppb) deemed of volcanic origin by Sigl et al. (2015). We also indicate the distinction made by these authors between "bipolar" volcanic signals (i.e., those found in ice-cores in both hemispheres and deemed to represent tropical or low-latitude eruptions) and those signals found only in Greenland ice (and deemed to represent extratropical Northern Hemispheric eruptions). Sigl et al. (2015) also provide an estimate of global radiative forcing (in terms of the cumulative annual reduction of solar energy reaching the Earth's surface, in watts per square meter) associated with the sulphate output from each eruption, and we now also cite these values in the main text.

Additionally, we plot negative growth values from the temperature-sensitive NH tree-ring chronology (N-Tree) employed by Sigl et al. (2015), which clearly shows the growing season temperature impacts of many of the eruptions experienced during the Western Han era (206 BCE - B CE). We further plot the precipitation data provided by the Qin et al. (2025) study highlighted by the reviewer below (where we further discuss this interesting record).

[Figure]

**R 2.4:** Figure 3 and the relevant text. Please provide brief description of how "vagrancy "and "planned migration" are identified. What criteria goes into the definition, the direction, distance, population scale, etc? And what implications do such difference have?

**A 2.4:** We thank the reviewer for these queries.

We have now explained these terms more explicitly in the main article text and are also happy to elaborate here. Thus, vagrancy refers to historical records where the word "vagrant" (流民) is directly used or where historical accounts clearly describe groups of people "abandoning their homes or land" (line 301 in our original submission; the location of the text may change after adjustments). Planned migration events, by contrast, refer to historical records that report people being relocated long-term (years or more) following edicts or administrative orders, etc. Escaping, fleeing, and surrender events during or due to war are not counted, because, based on a detailed examination of these records, their scale appears relatively small, and their impacts are temporary, unless historical sources clearly mention that the surrendered populations were arranged to be resettled in specific areas (e.g., the over 50,000 people led by Chanyu surrendered to the Han government in 55 BCE).The movement of populations for temporary labour needs is also not counted, as these populations were dismissed from their duties after a short period of time.

We also examined all the relevant records we found from historical sources with *The population and geography of Western Han (西汉人口地理)* (Ge, 2014), a major study which, although it does not directly provide a dataset, lists many records describing cases of population changes as mentioned in section 2.1.

For our SEA approach, we require only a chronology of the annual frequency of vagrancy and documented planned migrations. Thus, information regarding the direction, distance, and population scale does not play a part in our count. We have provided this information (if available from historical sources) in table S5 only for reference, because there is no scholarly compilation of population changes in the Western Han Dynasty currently available in Chinese or English. We believe that this can provide some convenience for researchers who study this topic in the future.

**R 2.5:** Section 3.2. The comparative case study could benefit from a more constructive and easy-to-follow structure, with a section summary highlighting the main conclusions from the comparison.

**A2.5:** See also our response A1.11. We have adjusted the discussion and conclusion sections addressing this comment and Reviewer 1's related comment R1.11.

**R 2.6**: Instead of being a stand along case investigation, the study could benefit from some parallel discussion on the climatic impacts of same eruptions in other society, such as Ptolemaic Egypt and the Ancient Near East (Line 185-188), or the climatic and social impacts of more recent eruptions in Imperial China.

**A2.6:** Reference to both suggested cases can be found in several parts of the main article text, with additional discussion now offered in the Discussion and Conclusion. For example, we highlight the case of Ptolemaic Egypt (using the studies of Manning et al., 2017, Ludlow and Manning, 2021, and Singh et al. 2013) with respect to the apparent link between explosive volcanism and internal revolt in Egypt (but also the cessation of inter-state warfare conducted by Egypt). We also cite Ludlow et al. (2023) in terms of motivations for state-planned migrations elsewhere in the ancient world, and newly now in terms of the apparent association between documented societal stresses (famine, conflict, etc.) and explosive volcanism between 750 and 650 BCE in the Ancient Near East. We also highlight the work of Gao et al. (2017) in examining the impact of the 1815 CE Tambora eruption on Qing dynasty China.

Gao, C., Gao, Y., Zhang, Q., and Shi, C.: Climatic aftermath of the 1815 Tambora eruption in China, Journal of Meteorological Research, 31, 28-38, 2017.

Ludlow, F. and Manning, J.: Volcanic eruptions, veiled suns, and Nile failure in Egyptian history: Integrating hydroclimate into understandings of historical change, in: Climate Change and Ancient Societies in Europe and the Near East: Diversity in Collapse and Resilience, Springer, 301-320, 2021.

Ludlow, F., Kostick, C., and Morris, C.: Climate, violence and ethnic conflict in the Ancient World, The Cambridge world history of genocide, 1, 2023.

Manning, J. G., Ludlow, F., Stine, A. R. Boos, W., Sigl, M. and Marlon, J: Volcanic Suppression of Nile Summer Flooding Triggers Revolt and Constrains Interstate Conflict in Ancient Egypt, Nature Communications, 8, Article 900, 2017.

Sigl, M., Winstrup, M., McConnell, J. R., Welten, K. C., Plunkett, G., Ludlow, F., Büntgen, U., Caffee, M., Chellman, N., and Dahl-Jensen, D.: Timing and climate forcing of volcanic eruptions for the past 2,500 years, Nature, 523, 543-549, 2015.

Singh, R., Tsigaridis, K., LeGrande, A. N., Ludlow, F., and Manning, J. G.: Investigating hydroclimatic impacts of the 168–158 BCE volcanic quartet and their relevance to the Nile River basin and Egyptian history, Climate of the Past, 19, 249-275, 2023. [Updating this from the previously cited Discussion paper].

**R 2.7:** In addition to the documentary data, the results and discussion could benefit (be more concrete and solid) from including proxy records such as tree-ring. For example, Qin et al (2025, copied below) provides annually resolved tree-ring records and process-based physiological modelling results of hydroclimate conditions of northern China during 270 to 77 BCE. It would be helpful to check out their results and compare them with results of this study.

Qin, B. Yang, A. Bräuning, etc, Persistent humid climate favored the Qin and Western Han Dynasties in China around 2,200 y ago, Proc. Natl. Acad. Sci. U.S.A. 122 (1) e2415294121, https://doi.org/10.1073/pnas.2415294121 (2025).

**A 2.7:** We thank the reviewer for highlighting this important study. We now cite it in the main article text and emphasize its finding of the beneficial effects of a more humid period (on average) on the fortunes of the Western Han dynasty. This also allows us to note that a society may still be vulnerable to sudden departures from a climatic norm, as can be induced by large explosive volcanic eruptions, such as those experienced by the Western Han.

We have also added the precipitation reconstruction offered by Qin et al. (2015) to the new figure described in A2.3 (above). This suggests a notable correspondence between drier years and several of these large eruptions. This is consistent with the finding in our SEA analysis that shows an association between documented droughts and eruptions. However, not all eruptions appear to coincide with drier conditions in the Qin et al. (2025) record. Understanding this requires further investigation, but we note that (1) it may partly arise from the seasonality of the human-documented droughts versus those identifiable in the tree-ring-based evidence, (2) the location of the tree-rings employed (from Jingyuan and Gansu) and how representative they may be of the regions of human-documented drought, and (3) the size and location of the volcanic eruptions in question (our sample during the Han period is too small to offer any conclusive study of whether tropical or NH extratropical or tropical eruptions are more clearly associated with drought (and for which regions across the vast landmass of China)). Moreover, the Qin et al. (2025) precipitation reconstruction currently ceases in 77 BCE, and so we cannot currently use it to examine the effect of the largest eruption of our period (in 43 BCE).

Nonetheless, even the brief consideration of the Qin et al. (2025) record usefully highlights remaining open questions about volcanic hydroclimatic impacts for China, and the potentially differing signals captured by human and natural climate proxies. We now note these issues in the main manuscript text.

---

## Author Response (AR2)

**Responses to Reviewer 1**

**General comments:**

The revised manuscript demonstrates substantial improvements, with the authors addressing reviewer feedback through enhanced historical contextualization, clearer data presentation, improved methodological clarity, more effective visual aids, and increased technical accuracy. Notable revisions include a stronger introduction to key historical figures, improved data visualizations, a more comprehensive literature review, refined analytical categorizations, and corrections to supplementary materials. These enhancements significantly elevate the manuscript's accessibility, scholarly rigor, and readability. With minor revisions, it is well positioned for publication and offers valuable insights into the volcanic impacts on the Western Han Dynasty.

We appreciate the reviewer's positive feedback regarding our revised manuscript and the constructive comments to help us improve.

**Specific comments:**

Line 392: It would be helpful to briefly explain the meaning of the term Chanyu to aid readers who may not be familiar with it. I also suggest reviewing the manuscript for other Chinese phrases or terms to see if similar clarifications might enhance accessibility for a broader audience.

Line 531: It is recommended to delete the term Fangshi, as readers who are not familiar with Chinese history may not understand its meaning.

**Technical corrections:**

Line 234: The correct spelling of the place name should be Jingyuan.

Line 576: There is an extra comma between the name and volume number in the citation of historical documents, which is inconsistent with the formatting used elsewhere in the manuscript. The same issue is observed in lines 578, 615, 637, 682, and 684.

The citations of figures: The citations of figures in the text are inconsistent. Some instances use the full term "Figure" while others use the abbreviation "Fig." It is recommended to standardize the format throughout the manuscript.

References: The formatting of the newly added references appears to be inconsistent with the existing ones. In particular, there should be a new line separating the two references on line 892.

Thank you for the careful reading and the comments; we have corrected all these matters as suggested.

**Responses to Reviewer 2**

The authors have sufficiently addressed the questions and suggestions raised during the previous review. I therefore recommend the acceptance of the manuscript, after the authors satisfactorily address the following issue:

Please consider rewrite the second half of the abstract, focusing on summarizing the results drawn from this investigation, instead of (or expanding on) the justification of choosing the two case study periods.

We appreciate the reviewer's positive feedback regarding our revised manuscript and the helpful suggestion. We have made changes accordingly.